

# Facial width-to-height ratio relates to dominance style in the genus *Macaca*

Marta Borgi[1,2] and Bonaventura Majolo[1]

[1] School of Psychology, University of Lincoln, Lincoln, United Kingdom
[2] Department of Cell Biology and Neurosciences, Istituto Superiore di Sanità, Rome, Italy

## ABSTRACT

**Background.** Physical, visual, chemical, and auditory cues signalling fighting ability have independently evolved in many animal taxa as a means to resolve conflicts without escalating to physical aggression. Facial width-to-height ratio (fWHR, i.e., the relative width to height of the face) has been associated with dominance-related phenotypes both in humans and in other primates. In humans, faces with a larger fWHR are perceived as more aggressive.

**Methods.** We examined fWHR variation among 11 species of the genus *Macaca*. Macaques have been grouped into four distinct categories, from despotic to tolerant, based on their female dominance style. Female dominance style is related to intra- and inter-sexual competition in both males and females and is the result of different evolutionary pressure across species. We used female dominance style as a proxy of intra-/inter-sexual competition to test the occurrence of correlated evolution between competitive regimes and dominance-related phenotypes. fWHR was calculated from 145 2D photographs of male and female adult macaques.

**Results.** We found no phylogenetic signal on the differences in fWHR across species in the two sexes. However, fWHR was greater, in females and males, in species characterised by despotic female dominance style than in tolerant species.

**Discussion.** Our results suggest that dominance-related phenotypes are related to differences in competitive regimes and intensity of inter- and intra-sexual selection across species.

Corresponding author
Marta Borgi, marta.borgi@iss.it

## INTRODUCTION

In animals, aggressive conflicts may bear significant fitness costs for the opponents (e.g., chronic stress, severe injuries, limited access to resource, *House, Landis & Umberson, 1988*), that can potentially outweigh the benefits of winning a conflict. Because of such potential costs, visual, chemical, tactile and auditory cues, as well as physical traits, have independently evolved in various taxa to signal the fighting ability of opponents and to allow animals to resolve conflicts without escalating to physical aggression (*Arnott & Elwood, 2009*; *Parker & Rubenstein, 1981*). For example, animals can maintain their dominance rank in a social group by signals of dominance/submission and displacements that do not involve overt aggression (*Preuschoft & Van Schaik, 2000*).

In humans, facial phenotype, specifically the facial width-to-height ratio (fWHR), is a cue of aggression, dominance and fighting ability: fWHR is positively related to the acquisition of status and resources, antisocial tendencies, dominance status and propensity to be aggressive (*Carré & McCormick, 2008*; *Carré, McCormick & Mondloch, 2009*; *Haselhuhn & Wong, 2012*; *Lefevre et al., 2014a*; *Sell et al., 2009*). Preliminary evidence on the association between circulating testosterone and fWHR in men (*Lefevre et al., 2013*), as well as the association between individual differences in amygdala reactivity, fWHR and self-reported aggression (*Carré, Murphy & Hariri, 2013*), suggests a possible path through which fWHR may have evolved to signal aggressive attitude. The association between facial phenotype and aggression might reflect the common influence of pubertal testosterone on cranial growth and the development of neural circuitry underlying aggressive behaviour (*Carré & McCormick, 2008*; *Carré, Murphy & Hariri, 2013*). High fWHR may thus work as a marker of "masculine" tendencies, in particular in species with a sex-biased frequency of aggression. Previous studies on humans indicate that the correlation between fWHR and dominance-related behaviours, as well as the modulator effect of fWHR on the relationship between amygdala reactivity and self-reported aggression, are either specific to or more robust in men (*Carré & McCormick, 2008*; *Carré, Murphy & Hariri, 2013*; *Geniole et al., 2014*; *Goetz et al., 2013*; *Haselhuhn & Wong, 2012*; *Stirrat & Perrett, 2010*; but see: *Lefevre et al., 2014a*).

Similarly to what happens in humans, non-human primates rely extensively on non-verbal communication and on a variety of facial displays to signal aggression/submission and to modulate social interactions (*Maestripieri, 1997*). Therefore, facial phenotypes, such as the fWHR, can have a homologous function and a shared phylogenetic history in non-human primates and in the human lineage. In support to this hypothesis, two recent studies have shown that, in brown capuchin monkeys (*Sapajus apella*) fWHR is positively related to alpha status and to a dominance-related personality trait (*Lefevre et al., 2014b*; *Wilson et al., 2014*).

To date, research on the relationship between fWHR and behaviour has rarely been conducted on non-humans and it has mostly focused on within-species variation. If fWHR is a signal of aggression and fighting abilities, we predict correlated evolution between fWHR and social traits at the species level. In other words, we predict different evolutionary pressure on fWHR depending on the species-specific differences in aggressiveness, competitive regime and dominance style. We tested this hypothesis in the genus *Macaca*, comprising 22 species with similar group composition (i.e., multimale-multifemale social groups) but differing in female dominance style and intra-group social relationships, according to a four-grade scale ranging from despotic (grade 1) to tolerant (grade 4) (*Balasubramaniam et al., 2012a*; *Balasubramaniam et al., 2012b*; *Thierry, 2000*; *Thierry et al., 2008*; *Thierry, Iwaniuk & Pellis, 2000*). Despotic, grade 1, species are characterised by steep linear dominance hierarchies among females, whereby the outcome of dyadic competitive interactions, as well as access to resources, strongly depend on the dominance rank of the contestants (*Balasubramaniam et al., 2012a*; *Balasubramaniam et al., 2012b*; *Thierry, 2000*). Competitive interactions can quickly escalate into overt aggression and result in injuries if a subordinate animal does not display submission to a dominant individual (*Thierry, Iwaniuk & Pellis, 2000*). In tolerant, grade 4, species, dominance

hierarchies are shallow among females, that is, the outcome of dyadic competitive interactions depends on context as much as it does on the dominance rank of the contestants (*Balasubramaniam et al., 2012a*; *Balasubramaniam et al., 2012b*; *Thierry, 2000*). Low-intensity aggressive interactions and counter-aggression are more frequent in tolerant species than in despotic species, but conflicts less frequently result in injuries, and access to resources is less rank-dependent (*Balasubramaniam et al., 2012a*; *Balasubramaniam et al., 2012b*; *Thierry, 2000*; *Thierry et al., 2008*; *Thierry, Iwaniuk & Pellis, 2000*). Species in-between the two extremes of this grading system (grade 2 and 3), show a mixture of dominance style traits of tolerant and despotic species (*Balasubramaniam et al., 2012a*). Differences in dominance style across macaque species have a strong phylogenetic signal (*Balasubramaniam et al., 2012a*; *Balasubramaniam et al., 2012b*).

In female macaques, we predict correlated evolution between dominance-related phenotypes and species-specific differences in dominance style: fWHR should be greater in females from despotic species than in those from tolerant species. In despotic species, displays of dominance, aggression and submission among females should be more important than in tolerant species, as in the former the risk of escalated aggression is expected to be higher. Over evolutionary time, therefore, there should have been stronger pressure for the evolution of dominance-related phenotypes in female macaques in despotic species than in tolerant species, other things being equal (relative dominance rank of the individuals). We predict a significant relationship between fWHR and female dominance style across species in male macaques. However, we cannot predict whether such relationship is positive or negative, as it is currently not known if male macaques show a similar or opposite pattern of inter-specific differences in dominance style as observed in females. Our prediction for male macaques is based on the fact that competitive regimes among females are related to male-male competition and dominance style (*Schülke & Ostner, 2008*; *Schülke & Ostner, 2012*) and can affect intra- and inter-sexual selection in both sexes. For example, male reproductive skew is more pronounced in female tolerant species (*Schülke & Ostner, 2008*; *Schülke & Ostner, 2012*; *Van Noordwijk & Van Schaik, 2004*). To test our predictions we used the four-grade scale of dominance style in female macaques as a proxy of differences across species in the level of inter- and intra-sexual selection in the two sexes. We calculated fWHR from 145 two-dimensional images (a measure previously shown to correlate with 3D scans and facial anthropometry, *Kramer, Jones & Ward, 2012*) of male/female faces of macaques from 11 species representing the four grades of female dominance style—as in *Thierry*'s classification (*2000*). We analysed the correlation between fWHR and female dominance style using phylogenetic-controlled analyses and standard (i.e., without phylogenetic control) multiple regression.

## MATERIALS & METHODS

### Images and measurements

We collected frontal images of the face of as many macaque species as possible following two different approaches. First, we used pictures taken by one of us (BM) and requested pictures taken by colleagues working on macaques in the wild or in captivity. Second, in

**PeerJ** ______________________________________________________

**Table 1  Images used for the analyses.** Number of images and mean fWHR (square root transformed) divided by species, sex and dominance style.

| Scientific name | Common name | Dominance style[a] | Number of pictures Female | Male | Total | Mean fWHR Female | Male |
|---|---|---|---|---|---|---|---|
| M. cyclopis | Formosan rock macaque | 1 | 2 | 4 | 6 | 1.18 | 1.10 |
| M. fuscata | Japanese macaque | 1 | 14 | 4 | 18 | 1.05 | 1.06 |
| M. mulatta | Rhesus macaque | 1 | 9 | 11 | 20 | 1.13 | 1.15 |
| M. nemestrina | Pig-tailed macaque | 2 | 2 | 12 | 14 | 1.04 | 1.05 |
| M. fascicularis | Long-tailed macaque | 2 | 4 | 4 | 8 | 1.10 | 1.16 |
| M. sinica | Toque macaque | 3 | 6 | 5 | 11 | 1.15 | 1.10 |
| M. arctoides | Stump-tailed macaque | 3 | 5 | 4 | 9 | 1.08 | 1.09 |
| M. sylvanus | Barbary macaque | 3 | 10 | 7 | 17 | 1.08 | 1.09 |
| M. radiata | Bonnet macaque | 3 | 8 | 5 | 13 | 1.09 | 1.07 |
| M. nigra | Crested black macaque | 4 | 6 | 7 | 13 | 0.99 | 0.97 |
| M. tonkeana | Tonkean macaque | 4 | 7 | 9 | 16 | 1.07 | 1.10 |
|  |  |  | 73 | 72 | 145 |  |  |

**Notes.**

[a] Category #1 defines female despotic species—grade #1 in Thierry's classification (*Thierry, 2000*)—and category #4 defines female tolerant species—Grade #4 in Thierry's classification.

order to expand our sample size in terms of number of pictures and species, we searched for images on the Google Images web search engine by submitting the scientific name of each species as a key-word. In order to be included in our dataset, images had to be frontal, full-faced photographs depicting male and female adult macaques (i.e., $\geq$ 5 years old for females and $\geq$ 7 years old for males) with a neutral expression (closed mouth) and with a resolution of at least 400 $\times$ 300 pixels per inch. For the images collected from the web, information on the sex and age of the animal in the image was often not available (sex and age of the animals was known for images taken by BM or obtained from colleagues). Therefore, these images were independently scored for age and sex by one of us (BM) and an expert on macaques (who was unaware of the aims of the study). Correlation between the two scores was positive and significant (Spearman correlation; age rho $=$ 0.70, $p < 0.001$; sex: rho $= 0.57$, $p < 0.001$). However, we discarded all of the images for which scores for sex and/or age were in disagreement between the two scorers in order to avoid biasing the analyses due to incorrect data on the sex/age of the animals. Moreover, since macaques often present individual physical traits which are easily identifiable, all images taken from the web were checked for independence by two researchers, in order to reduce the risk that the same animal could be depicted in two or more images included in the dataset (if that was the case, we only kept one image, with the highest resolution). Following this procedure, our dataset comprised a total of 145 images from 11 species of genus *Macaca*, 73 images of adult female macaques (40 images collected from the web) and 72 of adult male macaques (41 images collected from the web; Table 1).

Using Adobe Photoshop (Adobe Systems, San Jose, CA, USA), pictures were digitized at 72 dpi and were two-dimensionally rotated (horizontally aligned) and scaled to the same inter-pupillary distance, in order to standardise face size and head position across images.

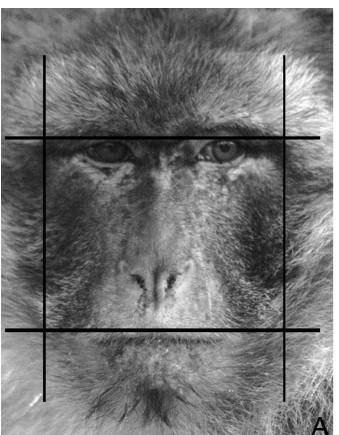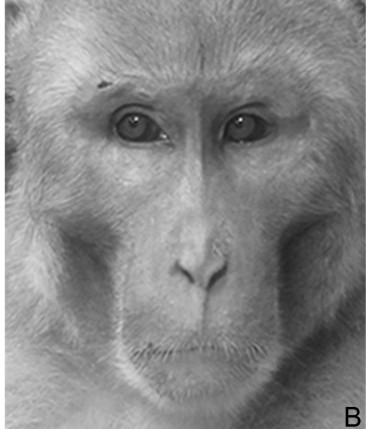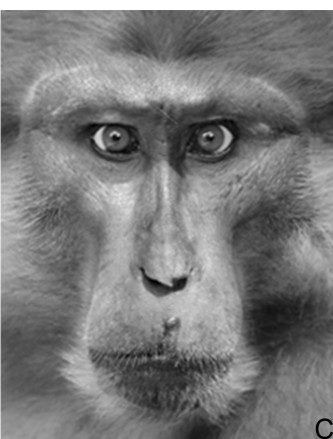

**Figure 1 Macaque faces.** (A) An example illustrating how fWHR was calculated from images (Barbary macaque). Bizygomatic width was measured as the horizontal distance between the left and right zygion (vertical lines); upper-face height as the vertical distance between the highest point of the eyelids and the highest point of the upper lip (horizontal lines). The fWHR was calculated as width divided by height. (B) A male rhesus macaque (dominance style 1). Photo by Lauren Brent (modified). (C) A male Tonkean macaque (dominance style 4). Photo by Bernard Thierry (modified).

Facial measurements were taken using the ruler tool with pixels as a unit. In accordance with previous work (*Kramer, Jones & Ward, 2012*; *Lefevre et al., 2014b*; *Wilson et al., 2014*), fWHR was calculated by dividing the bi-zygomatic width (maximum horizontal distance from the left to the right zygion) by the upper-face height (vertical distance from the highest point of the eyelids to the upper-lip) (Fig. 1). One of us (MB) took all of the measures; at the time of taking the measures MB was blind to the dominance style of the different macaque species in the dataset. In order to test the consistency of the facial measures taken across time, MB re-took the measures for the bi-zygomatic width and the upper-face height for a subset of images ($N = 39$), randomly selected for each species (2.4 images per species), six months after the same measures were first taken on those images. Consistency across the two measures was very high (Spearman correlation; bizygomatic width: rho = 0.93, $p < 0.001$; upper-face height: rho = 0.99, $p < 0.001$).

## Data analysis

Each species included in our dataset was assigned to one of the four grades of dominance style (from 1 = despotic up to 4 = tolerant species; Table 1) following the classification of macaque species available in *Thierry, 2000* (Table 6.2, p.112). We square-root transformed fWHR (dependent variable) to improve normality and used two distinct analytical approaches to test our hypothesis. First, we averaged the square-root transformed fWHR values per species and sex and ran a phylogenetically controlled generalized least square regression model (PGLS) independently for male and female macaques. Since the number of images available on each species and sex varied significantly in our dataset (Table 1), we entered the number of images for females (or males) as a control variable in the two PGLSs. We estimated Pagel's lambda (*Pagel, 1999*) using maximum likelihood. In the results we present the estimated lambda for each model and the p values of the likelihood ratio tests

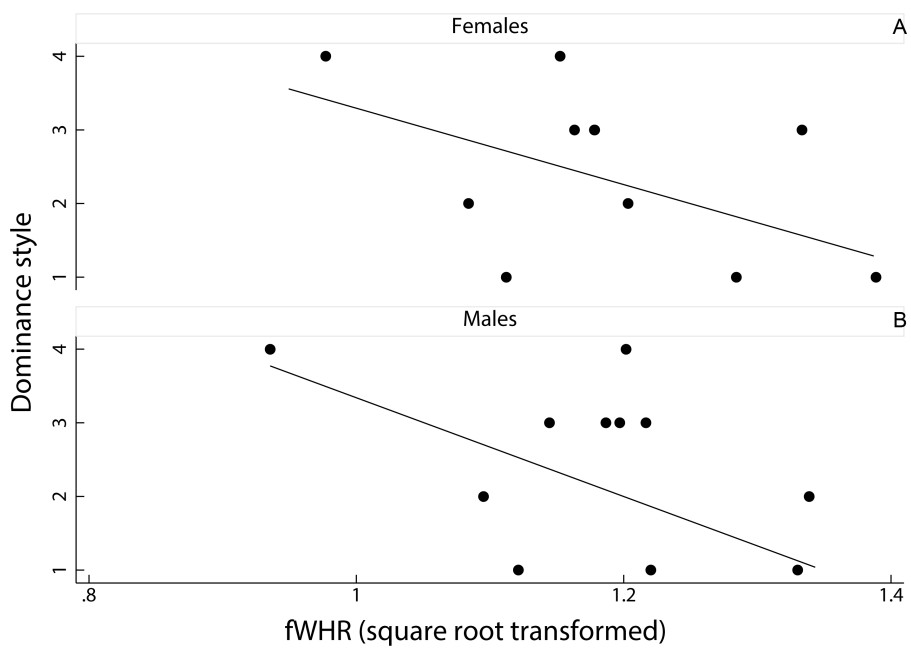

**Figure 2** **Relationship between female dominance style and fWHR (data averaged per species).** Scatter plot and line of best fit for the relationship between dominance style ($y$ axis) and fWHR ($x$ axis) in female (A) and male (B); grade 1 defines despotic species and grade 4 tolerant species (*Thierry, 2000*).

comparing the estimated lambda to the upper (lambda = 1) and lower bounds (lambda = 0). We ran these two PGLSs in R 3.1.0 (*R Development Core Team, 2011*), using the CAPER 0.4 package (*Orme et al., 2012*). We downloaded a consensus phylogenetic tree, with the chronogram branch option using the Genbank taxonomy, from the 10ktree primate phylogeny version 2 (http://10ktrees.fas.harvard.edu, *Arnold, Matthews & Nunn, 2010*). Second, since no phylogenetic signal was detected in the two PGLSs (see 'Results'), we ran a standard (i.e., without phylogenetic control) linear mixed model on the two sexes together. The fWHR values of each image (our dependent variable) were our data points ($N = 145$); dominance style and sex (females or males) were entered as fixed factors, species ID was our random factor. We ran this linear mixed model using Stata v.12.1 (*Stata Corp, 2011*).

## RESULTS

We had no significant result for the predicting variables in the two PGLSs run separately on females and males; the two full models were also not significant (PGLS on females: $F(2, 8) = 2.01$, adjusted R-squared = 0.17, $N = 11$, $p = 0.20$; PGLS on males: $F(2, 8) = 3.21$, adjusted R-squared = 0.31, $N = 11$, $p = 0.10$). Dominance style was negatively related to fWHR in the two sexes (i.e., fWHR was greater in despotic than in tolerant species) but this relationship was not significant (females: coefficient $\pm$ SE = $-0.02 \pm 0.01$, $t = -1.69$, $p = 0.13$; males: coefficient $\pm$ SE = $-0.02 \pm 0.01$, $t = -1.71$, $p = 0.13$; Fig. 2). The number of images available for each species and sex did not have a significant effect in any of the two PGLSs (females: coefficient $\pm$ SE = $-0.01 \pm 0.01$, $t = -1.20$, $p = 0.26$; males: coefficient $\pm$ SE = $0.01 \pm 0.01$, $t = 1.19$, $p = 0.09$). Pagel's lambda values (*Pagel, 1999*) were equal to

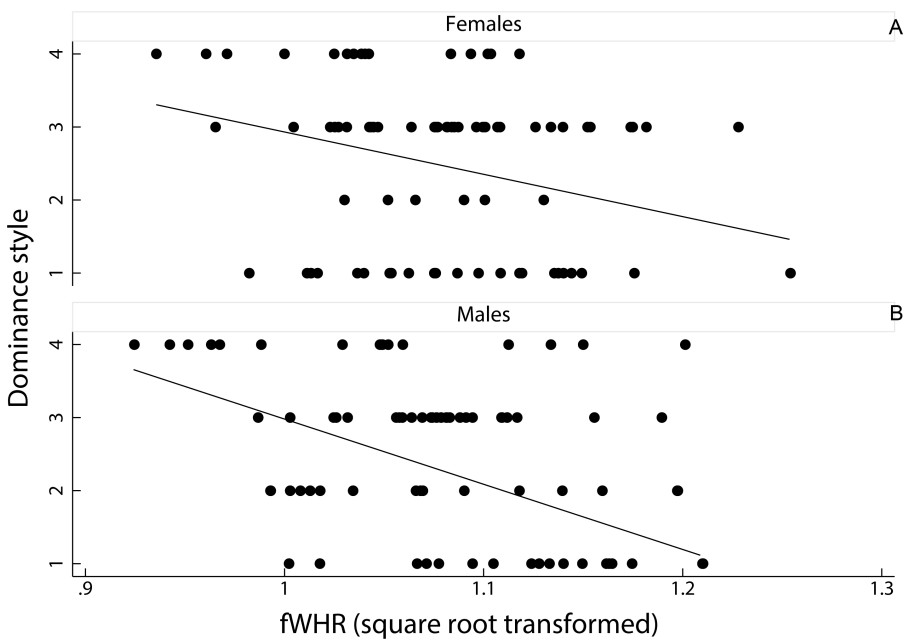

**Figure 3 Relationship between female dominance style and fWHR (data points represent each image in the dataset).** Scatter plot and line of best fit for the relationship between dominance style (*y* axis) and fWHR (*x* axis) in female (A) and male (B); grade 1 defines despotic species and grade 4 tolerant species (*Thierry, 2000*).

zero in the two PGLSs on males (likelihood ratio test: upper bound $p = 1.00$; lower bound $p = 0.12$) and females (upper bound $p = 1.00$; lower bound $p < 0.01$), indicating a weak phylogenetic signal for the relationship between fWHR and female dominance style.

A linear mixed model on the effect of female dominance style and sex of the animal, as fixed factors, on fWHR (species ID entered as a random factor) showed that fWHR was significantly greater in despotic species than in tolerant species (coefficient $\pm$ SE $= -0.20 \pm 0.01$, $z = -1.90$, $p < 0.05$; Fig. 3). However, fHWR did not significantly differ between female and male macaques (coefficient $\pm$ SE $= -0.00 \pm 0.01$, $z = -0.09$, $p = 0.93$).

## DISCUSSION

Our study is the first to analyse the relationship between species-specific differences in dominance style and fWHR. The four-grade scale of female dominance style was significantly related to fWHR (female despotic species having greater fWHR than tolerant species) in male and female macaques, but not when fWHR was averaged per species and sex. Moreover, we found no phylogenetic signal for differences in fWHR across macaque species.

fWHR is a signal of aggression and fighting ability that might facilitate the resolution of conflicts of interest without the need for these to escalate into overt aggression (*Carré & McCormick, 2008*; *Carré, McCormick & Mondloch, 2009*; *Lefevre et al., 2014a*; *Sell et al., 2009*). As such, fWHR may affect decision-making in competitive interactions (e.g., fight or flight) and can minimize the costs of competition to contestants for both won/lost and unresolved conflicts, including chronic stress, severe injuries and deaths (*Arnott & Elwood,*

*2009*; *Blanchard et al., 2011*). In humans, fWHR predicts aggressive behaviour (e.g., *Carré & McCormick, 2008*; *Goetz et al., 2013*), especially in males, and it may operate as a signal of physical dominance evolved under sexual selection (*Weston et al., 2004*; *Weston, Friday & Lio, 2007*). Our study suggests that fWHR may have a similar function and be related to sexual selection and sexual dimorphism in non-human primates (*Lefevre et al., 2014b*; *Leutenegger & Kelly, 1977*; *Plavcan & Van Schaik, 1992*; *Wilson et al., 2014*).

Analyses on female macaques support our prediction that species-specific differences in dominance style are related to fWHR, possibly because physical and behavioural traits have been under similar evolutionary pressure (*Balasubramaniam et al., 2012a*; *Balasubramaniam et al., 2012b*; *Thierry, 2000*; *Thierry et al., 2008*). We found a similar pattern in male macaques. However, the interpretation of our findings in males is difficult, since scarce data are available on dominance style difference across species in male macaques and on how such differences are related to female dominance style. Male reproductive skew is thought to be higher in species where female–female relationships are classified as being tolerant than in species where female–female relationships are despotic (*Schülke & Ostner, 2008*; *Schülke & Ostner, 2012*), other things being equal (e.g., operational sex ratio). Male reproductive skew is positively related to the degree of paternal relatedness in a species (*Ostner, Nunn & Schülke, 2008*; *Widdig, 2013*) but negatively related to female oestrous synchrony and mate choice (*Dubuc et al., 2011*; *Soltis et al., 1997*). Because of the limited data on male-male relationships, a parsimonious interpretation of our findings is that differences in female dominance style are related to species-specific differences in intra- and inter-sexual selection in the two sexes. Female dominance style can be used as a proxy of inter-specific differences in selection and competitive regimes. However, the exact nature of the cause–effect relationship between male fWHR and dominance style differences in males and females cannot be analysed until data are available. Because of the scarcity of data on males, since our results from phylogenetic analyses and multiple regression differed in their significance level (this could be due, at least partially, to the small number of species available in our dataset) and given that we cannot completely rule out the possibility that we had in the dataset more than one image for each animal (see 'Methods'), our findings have to be interpreted with caution.

Three areas of research require additional data and study testing alternative hypotheses. First, we know very little on the developmental trajectory of fWHR, the role of hormones, of the brain and of sexual maturation. In humans and other anthropoid primates, fWHR has been described as a sexually dimorphic trait that arises around puberty (coincident with the rise in pubertal testosterone) and that is, at least to some extent, not explained by sex differences in body size (*Carré & McCormick, 2008*; *Weston et al., 2004*; *Weston, Friday & Lio, 2007*). Preliminary evidence has shown a positive correlation between fWHR and circulating testosterone in men (*Lefevre et al., 2013*). Moreover, the association between fWHR and dominance-related behaviour may be more evident among individuals low in status (*Carré, 2014*; *Goetz et al., 2013*; *Welker, Goetz & Carré, 2015*). If so, life-history variables such as age at sexual maturation, degree of social instability (e.g., frequency of rank reversals) and of stress (due to competition, e.g., *Crockford et al., 2008*) could interplay with dominance rank in affecting fWHR throughout an animal's life.

Second, intra-species variance across populations/groups in dominance style is expected to be high but the causes of behavioural flexibility are still little understood (*Kamilar & Baden, 2014*). The scarcity of images of macaque faces for which data were also available on the dominance rank of each animal, as well as on individual- and group-specific social traits (e.g., conciliatory tendency, steepness of the hierarchy or frequency of counter-aggression), forced us to enter in the analyses the four-grade system of dominance style instead of more specific measures of social style. Ideally, additional data are needed to analyse the relative role of each social trait contributing to the species/population/individual dominance style, the degree of inter- and intra-species variation in fWHR, and to what extent such variation is explained by shared phylogenetic history or current socio-ecological factors (e.g., level of competition in a group).

A third limitation of our study, and an area that requires further investigation, is whether two-dimensional images used to calculate facial measurements reliably 'represent' how non-human primates see faces of their conspecifics, especially in the case of pictures taken in the absence of controlled conditions.

## CONCLUSIONS

In conclusion, our research makes a novel contribution to the study of dominance-related phenotypes, by showing that fWHR is related with female dominance style in male and female macaques. This study has to be considered as one of the first steps towards understanding whether and how sexual selection, socio-ecological variables, reproductive strategies and life-history variables affect dominance-related phenotypes.

## ACKNOWLEDGEMENTS

We would like to thank Rebecca Ayre for help to collect the images used in this study. We are extremely grateful to Simone Giancontieri and Christopher Young for their help with the fWHR measures and data analysis, and to friends and colleagues who shared their images of macaques with us, in particular to Lauren Brent, Emily Bethell, Jerome Micheletta, Sandra Molesti, Bernard Thierry, the Parco Faunistico Piano dell'Abatino and the Caribbean Primate Research Center (supported by the National Center for Research Resources and the Office of Research Infrastructure Programs of the National Institutes of Health, grant 8-P40 OD012217-25). We would like to thank James Higham, Juliane Kaminski, Jerome Micheletta and Gabriele Schino for useful comments on previous drafts of our manuscript.

### Funding
The authors received no funding for this work.

### Competing Interests
The authors declare there are no competing interests.

## Author Contributions

- Marta Borgi conceived and designed the experiments, performed the experiments, wrote the paper, prepared figures and/or tables, reviewed drafts of the paper.
- Bonaventura Majolo conceived and designed the experiments, analyzed the data, contributed reagents/materials/analysis tools, wrote the paper, prepared figures and/or tables, reviewed drafts of the paper.

## Data Availability

Raw data can be found in the Supplemental Information

## Supplemental Information

Supplemental information for this article can be found online at http://dx.doi.org/10.7717/peerj.1775#supplemental-information.

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
