# Peer review of "Facial width-to-height ratio relates to dominance style in the genus Macaca"

_PeerJ, doi:10.7717/peerj.1775_

## Round 0.1 · original submission · Major Revisions

As you will see both reviewers like your manuscript but have some major comments which you need to address. Here I would like to especially highlight the comment about sample size, which both reviewers found insufficient. It would be great if you could increase the sample size especially for the species for which access to materials should not be very difficult.

·

Basic reporting

The paper is nicely written and contains all the information necessary to understand the work. The structure is adequate and the figures useful. It would be nice to see an additional figure, showing the faces of the different species (or 1 exemplar per dominance grade) to illustrate the finding.

Experimental design

The design of the study is clear and straightforward. The research question is clearly defined and addresses an aspect of primate behaviour that has received little attention so far. The methods used seem appropriate, and are certainly described with the required level of detail to be reproduced, and with appropriate reference.

Validity of the findings

I am not an expert in PGLS analysis but it seems to have been conducted appropriately.

I am a bit surprised by the limited sample size for some species. Although I totally understand that it can be hard or even impossible to find high quality photographs of rare and understudied macaque species (e.g. Moor or Booted macaques), I would have thought that it would be relatively easy to find more photographs of the most common or well-studied species (e.g. rhesus or Japanese macaques). If this is not possible, could the authors add an additional analysis, restricted to a subset of the data where at least more than 1 photograph is used (assuming that this covers the 4 grades and is enough for the PGLS analysis)?

Similarly, it would have been interesting if more images came from researchers able to provide specific information about the particular groups where the images were taken. Then, it would be possible to use more precise estimates of social styles that would be specific to these groups (e.g. conciliatory tendency, counter-aggression rate). Could this be done on the subset of data provided by researchers?

Even if conciliatory tendencies and/or rate of counter-aggression specific to particular group cannot be used, you could use an average per species rather that the 4-grade scale. This would give a more precise picture of the relationship between fWHR and dominance style (see Dobson, S. D. (2012). Coevolution of Facial Expression and Social Tolerance in Macaques. American Journal of Primatology, 74(3), 229–235. http://doi.org/10.1002/ajp.21991 for a similar approach with repertoire size).

Additional comments

l.43: add ')' after '1988)'.

l.100: add ',' after 'former'.

l.104-106: I can see how this prediction makes sense in the general context of sexual selection, but not really based on males being dominant over females. Could you amend this or develop to explain?

l.108-111: The sentence does not seem complete. Could you add a description of what was done with the data to test the prediction (e.g. Then, we analysed the relationship between fWHR and dominance style using phylogenetically informed correlations)?

l.172: add ')' after '0.47'.

·

Basic reporting

This is a short and well written manuscript investigating the question of facial width to height ratios in macaques.

Experimental design

I have 3 issues related to the experimental design:

1) I think that some species should be removed or the sample size increased. I think a species should have a minimum sample size of 2 individuals per sex to be included. A sample size of 1 individual as representative of any one sex is insufficient, as that 1 individual may be atypical. I also think that including species in the analysis of one sex and not the other (as indicated in Table 1) is problematic, especially when you’re concluding that things are different for males and females. You might have a different result for males and females solely because of the inclusion of different species in the analysis.
2) I don’t think you should be running PGLS separately for males and females – run one model with all your data and with Sex and Sex*Social style (or whatever, see comment #4 below) in the model.
3) How can you be sure that the individuals in your sample are independent? Tourists posting pics on the internet have often seen and photographed the same few animals at a sanctuary or zoo.

Validity of the findings

No comments.

Additional comments

4) There seems to be a major conceptual and framing problem with the manuscript. Thierry’s social styles, which the Authors use to categorize their species, are largely based on data collected on female-female relationships, not on male-male relationships. They are appropriate for considering females, but not males, and it remains unclear whether male-male social relationships in these species mirror those of females. For example, while studies of captive Tonkean macaques show that social style patterns are similar in males to females (Thierry 1985; Thierry et al. 1994), a study of wild stump-tailed macaques found that males show less tolerance towards other males than females show towards other females (Richter et al. 2009). When it comes to males, it is also critical to note that there is an inverse relationship between Thierry’s social style categories (used in the present study), and the extent to which males compete with other males over reproduction and dominance. Species categorized by Thierry as socially tolerant are actually those with the steepest reproductive skew by male dominance (Schulke and Ostner 2008 AJP; Schulke and Ostner 2013 chapter in The Evolution of Primate Societies book), exhibiting the most marked body and canine size sexual dimorphism (Plavcan 2004), and exhibiting the strongest direct male-male competition over dominance rank (van Noordwijk and van Schaik 2004, chapter in Sexual Selection in Primates book). In fact, males living in tolerant societies according to the Thierry scheme are defined as being despotic according to the male-male coalition literature (Pandit and van Schaik 2003, BES), based on the high reproductive skew (i.e. little “sharing” of reproduction), while males in societies defined as despotic by Thierry are defined as egalitarian in the male-male coalition literature (Pandit and van Schaik 2003), based on their low reproductive skew (i.e. greater “sharing” of reproduction). One could easily frame your male result as one related to male-male competitive regime, degree of reproductive skew (high for the tolerant species, low for the despotic species, see Schulke and Ostner 2008, 2013), and mode of dominance acquisition (van Noordwijk and van Schaik 2004). According to your data, macaque species with more direct contest competition and more marked sexual dimorphism have reduced fWHR relative to those species with less direct contest competition and lower sexual dimorphism. Not including factors such as degree of sexual dimorphism in your analysis seems highly problematic given this. Collectively, these points seem a major conceptual problem for your manuscript, and they need addressing, regardless of where the MS is published.

---

## Round 0.2 · Major Revisions

As you will see both reviewers think your manuscript has improved but especially reviewer 2 still has comments which I would ask you to respond to.

·

Basic reporting

No Comments

Experimental design

No Comments

Validity of the findings

No Comments

Additional comments

The authors have addressed my previous comments. They have added more images and excluded species for which they did not have a sufficient sample size. The authors have also added a new analysis given the lack of phylogenetic signal. The conclusions are reasonable given the results, and the authors recommend caution when interpreting some of these.

I think this paper will stimulate more research and the authors are highlighting some interesting avenues for future studies. I am happy to recommend this manuscript for publication.

There are only a couple of minor points that might require the authors attention. These are detailed below:

l.102-109: For females, the authors could be more explicit about the prediction after presenting their reasoning (i.e. they expect fWHR to be higher in females from despotic species than in tolerant one; or a significant negative relationship).

l.186: Since dominance style was negatively related to fWHR in the two sexes, the sentence between brackets could be ‘fWHR was greater in despotic than in tolerant species’ or it should include a statement about males.

l.195: Here again, if I am not mistaken this result applies to both sexes (the effect of female dominance style is reported) not females only.

·

Basic reporting

The manuscript is conceptually improved, and remains interesting. I still think there are methodological/analytical problems with the current version (see below). In addition:

1) The counter-argument against you having images of the same individuals that you’ve pulled from the web (mainly that the two observers don’t find this in their assessments of each image) should be included in the paper, not just the response. (Unless it’s there and I missed it?)
2) Note sure why you’re citing the original Schulke and Ostner 2008 paper but not the 2011 book chapter analysis, which includes more species?
3) Lines 223-228. You should also mention sexual dimorphism in body size and canine size here – it’s important given the paper content.

Experimental design

4) The main result rests on there being no phylogenetic signal, such that a standard MLR can be used. Only then are there significant results. However, it’s not sufficient just to find and say that lambda is 0, you also need to look at the probability function of lamda, and it’s not clear to me whether this has been done. PGLS might tell you that lamda is 0, but plotting the model estimate of lamda from 0 to 1 (x axis) against the probability of that lamda value (y axis) is required. You might find that any value from 0 to 1 is almost equally likely (ie. a broad flat curve with relatively equal probabilities across lamda values). This is especially likely with small sample sizes (few taxa). To use standard MLR you should know and show that lambda being 0 is far more likely than it being higher values.
5) Regarding your comment about use of multiple values per species (so that one model with both sexes could be run), you’re right that you shouldn’t do that in PGLS. An option would be to build a GLMM where one of the random effects is the variance-covariance matrix representing the phylogenetic control, and the other “Species ID” to account for the two measures per species. I’m not insisting you do this, but it’s worth looking at, and might give you cleaner and clearer results and a better paper. You can do it easily enough in MCMCglmm. I think also in ADMB.
6) I’m not sure why you’re using MLR rather than LMM? Isn’t the presence of Species ID as a term really there to control for the repeated measure? If so wouldn’t it be better to run an LMM. Response – fWHR; Fixed effects - Sex and Dominance Style; Random effect - Species ID.
7) I generally feel that the sample size is still low. There are an awful lot of people out there who might have good images of macaques. From the acknowledgements, few of these seem to have been contacted with requests for images.

Validity of the findings

See above regarding no requirement for phylogenetic control. A finding that lamda is equally likely to be high as low would make the use of standard (non-phylogenetically controlled) MLR invalid.

Additional comments

Apologies for apparently remaining anonymous on the previous review – this was unintentional.

I hope you find the above helpful - James Higham

---

## Round 0.3 · accepted · Accept

You have addressed the reviewers comments and your revision is now acceptable for publication. This is a really interesting study which I am sure will spark a lot of future work.